# Transfer of Mental Health Services for Medical Students to Cyberspace during the COVID-19 Pandemic: Service Use and Students’ Preferences for Psychological Self-Help Techniques

**DOI:** 10.3390/ijerph192013338

**Published:** 2022-10-16

**Authors:** Barnabás Oláh, Ildikó Kuritárné Szabó, Karolina Kósa

**Affiliations:** 1Department of Behavioural Sciences, Faculty of Medicine, University of Debrecen, 4032 Debrecen, Hungary; 2Doctoral School of Health Sciences, University of Debrecen, 4032 Debrecen, Hungary

**Keywords:** COVID-19, medical students, mental health, psychological self-help techniques

## Abstract

The high risk of mental health problems among medical students has been compounded by the COVID-19 pandemic, which greatly reduced social contact. The mental health support service of the medical school of one Hungarian university was transferred to the online learning management system and was expanded by self-help materials in three domains: Improving study skills, stress management techniques, and reducing stress related to the COVID-19 pandemic. We wanted to understand the preferences of medical students for psychological self-help techniques by investigating the pattern of access to online self-help materials and the characteristics of the users. Access to the online materials between April 2020 and April 2021 among Hungarian and international medical students was analyzed using the logging data of the system. Of all the students who logged in during the examination period (*n* = 458), 36.6–40.4% viewed materials to improve study skills and 23–29% viewed stress management materials, of which short-duration audio format techniques were preferred. The access rate of content targeting coping with the mental health effects of COVID-19 was 9.5–24%. Support to improve study skills is significantly more preferred than interventions targeting distress-reduction. The pattern of access can be used for the development of interventions that are of most interest to medical students.

## 1. Introduction

Medical students have been at a high risk of psychological distress and mental health problems [1,2,3]. The summary estimate of the prevalence of depression or depressive symptoms was found to be 27.2% in a systematic review of 183 studies involving more than 120,000 students from 43 countries [4]. 

However, there has been a dearth of consistent findings on the effectiveness on setting-based mental health-promoting interventions among medical students. In addition, available data have quality problems [5]. The internet and new digital technologies have been increasingly used by students and are reportedly important sources of mental health information and support but discerning between trustworthy and unreliable sources as well as concerns about personal data protection have been unsettled issues for many students [6]. Internet-based self-help materials can be considered cost-effective means for mental health promotion and might suit students that are not willing to seek face-to-face help due to fear of stigmatization or embarrassment [7]. Further development of such services requires the involvement of stakeholders and situation experts at an early stage [8]. Qualitative research offers the opportunity to obtain in-depth information, but data collection is usually time-consuming, and results are difficult to generalize [9].

Another approach is the evaluation of existing online services to describe the patterns of use and to identify the most accessed materials. This is relatively easy to do because the use of online services is tracked by the system itself, thus providing highly reliable data on usage. However, to calculate the rate of access requires information about the total number of potential users, which is information that is lacking in surveys in which online materials are available to an unspecified number of persons. Our study aimed to evaluate online mental health services based on precise data on potential and actual users during one year of the COVID-19 pandemic.

A large number of papers investigated the mental health status of university students during the COVID-19 pandemic. An online survey was carried out at Texas A&M University in the early stages of the pandemic (May 2020) and involved 2031 students, 48.14% of whom showed a moderate-to-severe level of depression, 38.48% showed a moderate-to-severe level of anxiety, and 18.04% had suicidal thoughts. Almost three-quarters of the students indicated that their stress/anxiety levels had increased during the pandemic [10]. A systematic review of papers on the mental health of Bangladeshi university students conducted between April–May 2020 indicated that prevalence rates of mild to severe symptoms of depression, anxiety and stress ranged from 46.92% to 82.4%, 26.6% to 96.82%, and 28.5% to 70.1% [11]. Mental health problems provoked by the pandemic were prevalent among Bangladeshi university students, and COVID-19 worry predicted anxiety, depression, and poor mental health status [12]. Mental health surveys carried out before and after the beginning of the pandemic in the US [13] and Italy [14], suggesting that the COVID-19 pandemic did cause persistent changes and aggravated mental illness symptoms among university students. The prevalence of moderate–severe anxiety increased by 7.2% and the prevalence of moderate–severe depression increased by 10.2% within four months after the pandemic began in a sample of 419 US first-year students [15].

The most frequently used strategies to cope with COVID-19 by Polish students from 17 universities were acceptance, planning, and seeking emotional support that was less adequate in younger students [16]. Approximately half of the students were found, in other studies, to be in need of help to cope with the situation created by the pandemic [10]. 

The mental health of medical students has been even more concerning compared to their same-aged peers in non-medical courses [17]. The COVID-19 pandemic compounded the mental burden inherent in the medical educational environment [18], while social distancing rules reduced the available support [19]. This necessitated the development of novel, online support services around the world [20], including in Hungary [18,19,20]. Mental health support providing individual consultation opportunities for medical students at the University of Debrecen was transferred to the online learning management system of the University from 15 April 2020 onto a dedicated portal. In addition to moving this service online, a wide range of self-help materials were developed and uploaded to the same portal. This portal was created as a special e-learning course with no pre-conditions for access and no requirement for completion within the learning management system of the university. Two courses (one in Hungarian and one in English) were created for medical students in the Hungarian and English courses and both provided similar services. The management system of the portal automatically tracked access to the materials. 

In line with the recommendations made early in the pandemic to research its mental health consequences and potential interventions [21], and considering the dearth of research on the patterns of use of e-mental health services [6], by medical students, we aimed to investigate access to our newly developed, closed-access online mental health support service and the characteristics of those who used the service between 15 April 2020 and 15 April 2021 based on data obtained from the system.

## 2. Materials and Methods

### 2.1. Description of the Online Form of Individual Consultation

Personal counseling previously had to be requested by writing an email to a specified address. After moving to the service online, interested students had to request a personal meeting by sending a message via the online portal. A dispatcher contacted the student within 24 h, providing the service email of a professional (qualified or in-training clinical psychologist) with whom a schedule was set up for individual consultation in the form of online video sessions instead of face-to-face meetings using a platform outside of the online learning management system of the university (outside of the portal at which the first contact was made). In November 2020, the method to request individual counseling was restored to the way it was before, thereby moving this service entirely out of the learning management system of the university. 

### 2.2. Description of the Newly Developed Online Mental Health Support Service

A domain specifically dedicated to mental health support was established in the online learning management system of the university, which students could access by logging in with their IDs. Brief self-help materials were developed in pdf, audio, video, or mobile application format in three domains: (1) Methods to improve study skills; (2) stress management techniques; and (3) education and advice related to the COVID-19 pandemic. For the first domain, a short presentation with plenty of examples was developed based on evidence-based effective study techniques, including suggestions for note-taking and time-management mobile applications. For stress-management, videos of varying durations were made available on deep breathing, relaxation, mindfulness meditation, autogenic training, yoga, as well as audio material for relaxation, and mHealth mobile applications. As to maintaining mental health and well-being specifically during the COVID-19 pandemic, evidence-based materials in written document format were made available.

### 2.3. Participants

All Hungarian (*n* = 1.282) and international (*n* = 1.707) medical students at the university received notification about the availability of the service via the administrative system of the university at the launch date, which was followed by two additional reminders. 

### 2.4. Data Collection

All activities on the portal were automatically tracked by the learning system from the point of entry until exit, thus enabling the monitoring of access to all materials on the portals. Since students had to use their university ID to log into the portal, data on course type, gender (by name), study year and type of study plan (recommended vs. individual) were also available. Data related to access were collected from the launch of the courses (15 April 2020) until 15 April 2021.

Ethics approval was issued by the Regional Institutional Research Ethics Committee, Clinical Center, University of Debrecen under the approval number RKEB_IKEB.5586A. By entering the learning management system of the University, the students consented to their activities being monitored. 

### 2.5. Statistical Analysis

Categorical variables were described by proportion and were compared using a chi-squared test. Continuous variables due to their non-normal distribution were described by median and interquartile range, as compared to the Mann-Whitney test. Statistical analysis was carried out by SPSS ver. 23 (IBM, Armonk, NY, USA).

## 3. Results

### 3.1. Visitors and Activity

14.3% (*n*= 183) of all Hungarian (female: 71.6%) and 16.1% (*n* = 275) of all international medical students (female: 58.9%) visited the website between 15 April 2020 and 15 April 2021. Of those, 53.6% (*n* = 98) of Hungarian and 50.5% (*n* = 139) of international students did not take further action, that is, they did not open any of the available materials. Most Hungarian students were in the first (18.6%) and fifth year (27.3%). 17.5% (*n* = 32) of them followed individual study plans due to one or more not completed exams. The majority of international students were in year one (24.4%) and three (20.7%), respectively. Among the international students, 19.3% (*n* = 53) followed individual study plans because of one or more uncompleted exams. There was no significant difference in the proportion of students at the Hungarian and international courses regarding the type of study plan (*p* = 0.836). Access was significantly higher among women (*p* = 0.006). A marginally significant difference was found between Hungarian and international students in terms of study year—a higher proportion of international students being in the first year of their study (*p* = 0.05). 

Of those internationals who followed individual study plans, 28% fewer students (37.7%, *n* = 20) viewed at least one document compared to those who followed the recommended study plan (52.3%, *n* = 116) (*p* = 0.058). The number of accessed materials per visitor was also lower among them (Mdn = 0, Q25 = 0, Q75 = 1), as compared to those following the recommended plan (Mdn = 1, Q25 = 0, Q75 = 1.25; Mann-Whitney U = 4900.50, *p* = 0.041). Demographic characteristics of the visiting students are presented in Table 1.

### 3.2. Preferences for Self-Help Resources

Figure 1 shows that 36.6% (*n* = 67) of Hungarian and 40.4% (*n* = 111) of international medical students accessed materials relating to improving study skills (*p* = 0.614), while 29% (*n* = 53) of Hungarians and 23.4% of internationals (*n* = 64) viewed content on stress-management techniques (*p* = 0.491), thus reflecting no significant difference between the two student populations. This was not the case for materials on COVID-19 related information, which 24% (*n* = 44) of Hungarian students opened, in comparison to 9.5% (*n* = 26) of international students, which made the group difference in access rate significant (*p* < 0.001). However, excluding one material, the Hungarian and the English language materials related to COVID-19 were not identical (Figure 2), which limited the comparability of access rates. In the case of the one identical material (a podcast on positive thoughts to cope with the quarantine), the difference in access rate was not significant among Hungarian and international medical students (*p* = 0.056).

Regarding access to materials on improving study skills, the material aimed at improving learning techniques received the most views in both groups, followed by time management and note-taking mobile applications (Figure 3). There were no significant differences between the two groups of students, and neither grade nor study plan showed any correlation with interest in different contents. In terms of gender, however, Hungarian women (11.5%, *n* = 15) were significantly more interested in organization help than men (1.9%, *n* = 1) (*p* = 0.040). This relationship was inverted in the international group, with males (13.3%, *n* = 15; females: 4.9%, *n* = 8) showing significantly greater interest (*p* = 0.014).

Investigating the formats of stress management materials, relaxation audios were the most accessed (Figure 4). Very brief relaxation audios with less than 2 min of duration had the highest access rate (16.4%, *n* = 30) among Hungarians, followed by mobile apps on stress management, specifically, those based on positive psychology (7.7%, *n* = 14) and cognitive behavioral therapy (CBT) techniques (7.1%, *n* = 13). All other stress management materials had an access rate of below 3% among Hungarian students. Medium length materials with a 5–15 min duration for relaxation with the same content as in Hungarian had the highest access rate (13.5%, *n* = 37) among international students, followed by videos on deep breathing (6.2%, *n* = 17). All other materials in English had access below 6%. Hungarian students showed significantly more interest compared to international students in mobile applications based on positive psychology techniques (*p* = 0.020) or CBT elements (*p* < 0.001).

Neither study year nor type of study plan were related to preference for stress management materials in any of the groups. Medium length relaxation audios were accessed by a significantly higher proportion of Hungarian males (15.4%, *n* = 8) than females (5.3%, *n* = 7) (*p* = 0.026). 4.9% (*n* = 8) of international female students accessed the yoga video, while international males showed no interest (0.0%) (*p* = 0.017).

Regarding the online form of individual consultation, no request arrived between April and November 2020 via the dedicated portal. In November 2020, the method of contact was changed, as described in the Methods Section. From then on, until April 2021, the number of students requesting individual consultation was similar to the mean number of students requesting consultation over the duration of the same time comparing it to the mean of the previous three years.

## 4. Discussion

Approximately one-sixth of all invited medical students accessed the online support services provided to medical students at the university during the first year of the COVID-19 pandemic. The view rates of our courses were similar to others found in other online health services. In a mixed sample of 308 community members and university students, only 3.6% of the respondents preferred self-directed online support; 10.7% preferred therapist-supported online interventions, and the majority (85.7%) indicated preference for face-to-face support [22]. Of the students, 10.1–13.8% had experience with telemental health resources in another sample from two U.S. Midwestern colleges (*n* = 662) [23]. A greater proportion of women exhibiting help-seeking behavior in our study also corresponds to the findings of others [24]. 

There was a relatively low interest in content specifically aimed at alleviating the psychological burden of the pandemic. This is probably explained by the fact that, by the time our services were launched (mid-April 2020), a wide selection of other sources of pandemic-related information were available, both within and outside the University. Mindfulness meditation, a popular technique [25], did not generate much interest in either group. Additionally, contrary to our expectations, international students showed little interest in mental health apps, in comparison to Hungarian students. In terms of distress-reduction methods, brief stress management techniques delivered in audio format were preferred, most liekly because of their convenient nature. Comparing the interest in distress-reduction methods with techniques to improve study skills, it can be concluded that medical students have a strong preference for interventions aimed at improving study skills, even during a pandemic crisis.

International students following individual study plans because of one or more noncompleted exams showed less interest in self-help materials in contrast to the supposition that noncompleted exams may be related to less effective study skills and/or stress-management techniques. Academic failure, coupled with a refusal to accept help, raises the possibility that this is indicative of a group of amotivated students. For these students, the optimal intervention could be effective career guidance in order to ensure that they choose a career according to their motivation and that state and university resources are better used and invested.

No request arrived for personal consultation in the first half year when first contact had to be made via the newly established portal, as opposed to the second half of the examined period when the number of requests was restored to the same level as before. This is most likely explained by the fact that students had to use their university ID to access individual counseling via the portal, which removed anonymity by identifying and logging students. This was probably discouraging, despite this step only being used to establish contact, and subsequent online consultation being taken outside of this system. This was clearly stated. Removing the first step (request for consultation) outside of the university system restored service use to the same level as before. The lack of anonymity as the most likely deterrent shows that anonymity is a crucial factor for medical students accessing personal psychological consultation. This is in line with the findings of a study on barriers of help-seeking among college students involving 13,984 first-year students in eight countries across the world [26]. The study measured a wide range of barriers to help-seeking, but only excessive self-reliance and being too embarrassed were independent predictors of reduced intent to seek treatment. In addition, depressive and anxiety symptoms showed the highest association with reporting embarrassment as a deterrent to seek help, as compared to other mental health issues. Since more than every fourth medical student has a high risk of depression [4], it is plausible that being embarrassed is the most a serious barrier to treatment in this population. 

One limitation of this study is the fact that all the students were aware of their activities being automatically registered on the portal; this may have been a deterrent for those students who did not want to reveal their help-seeking behavior. Another limitation is the fact that 15% of the invited students used the portal, but this was in line with other data regarding access to online resources, as described above. Generalization of the preferences for interventions to all medical students at the university is not possible due to the limited access, but it has been a well-known shortcoming of all preventive methods that target a wide population that many of those who would benefit from interventions do not feel the need to access them. An advantage of our study was its comprehensiveness in the sense that online services were provided to all medical students at the university without limitation. Our conclusions are based on actual access data taken from the log management system of the service website, which is a more exact measure of preferences than subjective self-reports that many other studies operate with.

## 5. Conclusions

Our results suggest that one size does not fit all. Mental help for medical students needs to be offered across a broad spectrum, in various formats and durations, as the uptake of different types of interventions may depend on gender and course. However, despite the fact that students showed a need for help to cope with the adverse mental health effects of the pandemic, they still preferred getting help in stress reduction in general (with a clear preference for relaxation techniques in short audio format), and most importantly in improving their study techniques. The outstanding preference for the latter is favorable and important in many ways, as their mental burden is mainly due to academic challenges [27]. Therefore, effective study-help interventions that are also utilized by the target population can reduce the need for stress management interventions. Thus, interventions to aid students in their studies seem to be a frequent need that requires problem-focused support. A variety of methods could be offered, such as training to improve learning skills, and should be part of the mandatory curriculum. Effective training should be delivered by trained professionals for small groups of students in the form of interactive exercises that monitor effectiveness. Peer mentoring can be established by recruiting senior students and pairing them with juniors. Voluntary peer mentoring programs cannot only help students to develop their study skills, but they can also help extend peer relations and integration into the community of medical students. These findings provide a reference for mental health interventions in this population and guide better use of resources for interventions.

## Figures and Tables

**Figure 1 ijerph-19-13338-f001:**
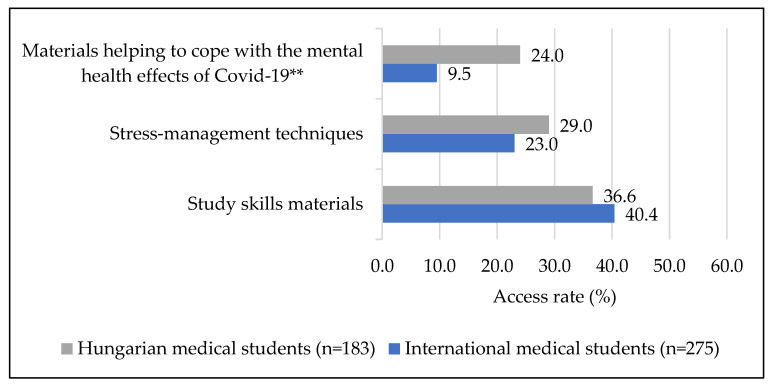
Access rate of the main content categories by course type; ******
*p*
**<** 0.001.

**Figure 2 ijerph-19-13338-f002:**
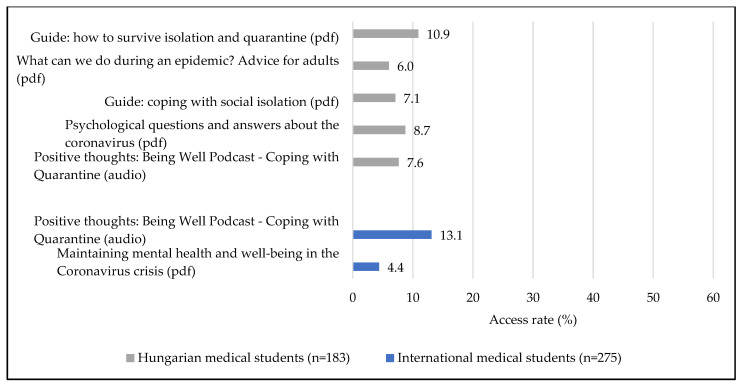
Access rate of materials helping to cope with mental health effects of COVID-19 separately by course type.

**Figure 3 ijerph-19-13338-f003:**
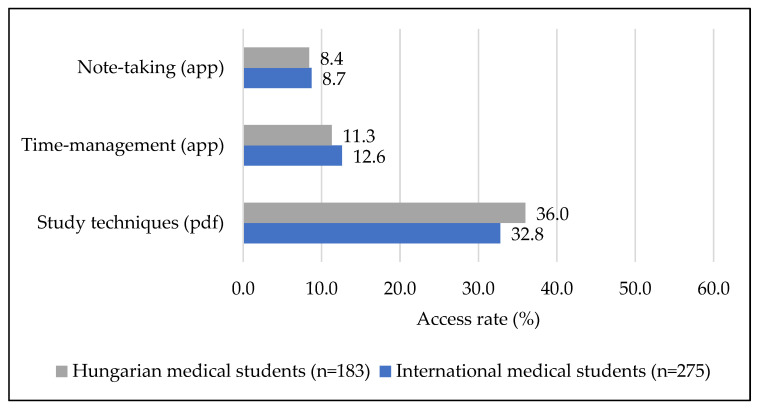
Access rate of study help contents separately by course type.

**Figure 4 ijerph-19-13338-f004:**
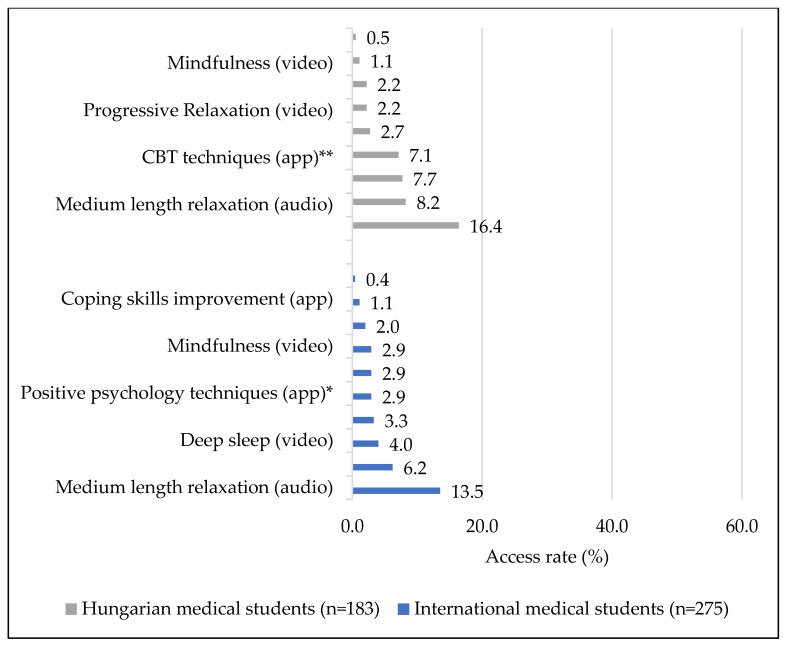
Access rate of stress-management materials separately by course type; * *p* < 0.05, ** *p* < 0.001.

**Table 1 ijerph-19-13338-t001:** Demographic characteristics of medical students using the services between 15 April 2020 and 15 April 2021.

	Hungarian Medical Students (*n* = 183)	International Medical Students (*n* = 275)	*p*-Value ^1^
Accessed any of the available materials N (%)	85 (46.4)	136 (49.5)	0.528
Gender N (%)			
Male	52 (28.4)	113 (41.1)	0.006 *
Female	131 (71.6)	162 (58.9)	
Year N (%)			
1st	34 (18.6)	67 (24.4)	0.050
2nd	20 (10.9)	36 (13.1)	
3rd	28 (15.3)	57 (20.7)	
4th	28 (15.3)	42 (15.3)	
5th	50 (27.3)	44 (16.0)	
6th	23 (12.6)	29 (10.5)	
Following individual study plan due to not completed exams N (%)	32 (17.5)	53 (19.3)	0.630

^1^ Pearson’s chi-squared test. * *p* < 0.05.

## Data Availability

The datasets used and/or analysed during the current study are available from the corresponding author on reasonable request.

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
