# Peer review of "Transfer of Mental Health Services for Medical Students to Cyberspace during the COVID-19 Pandemic: Service Use and Students’ Preferences for Psychological Self-Help Techniques"

_ijerph, 2022, doi:10.3390/ijerph192013338_

Round 1

Author Response

Dear Reviewer,

We would like to thank you for the careful reading of our manuscript and for your helpful comments. Please see our responses below.

R1.1. I recommend include an extensive review of the extant literature on the subject of COVID-19 and mental health, COVID-19 and stress, COVID-19 and coping, and other related issues. There is no in-depth literature review missing from the ‘Discussion’ section. Please do consider incorporating, for example, the following relevant studies:

BO: Thank you for the recommended studies. We have extended the review on COVID-19 incorporating more empirical studies as follows:

“A large number of papers investigated the mental health status of university students during the COVID-19 pandemic. An online survey was carried out at Texas A&M University in the early stage of the pandemic (May 2020) involving 2031 students 48.14% of whom showed a moderate-to-severe level of depression, 38.48% showed a moderate-to-severe level of anxiety, and 18.04% had suicidal thoughts. Almost three-quarters of the students indicated that their stress/anxiety levels had increased during the pandemic [10].  A systematic review of papers on the mental health of Bangladeshi university students conducted between April-May 2020 indicated that prevalence rates of mild to severe symptoms of depression, anxiety and stress ranged from 46.92% to 82.4%, 26.6% to 96.82%, and 28.5% to 70.1% [11]. Mental health problems provoked by the pandemic were prevalent among Bangladeshi university students, and COVID-19 worry predicted anxiety, depression, and poor mental health status [12]. Mental health surveys carried out before and after the beginning of the pandemic in the US [13] and Italy [14] suggested that the COVID-19 pandemic did cause persistent changes and aggravated mental illness symptoms among university students. The prevalence of moderate-severe anxiety increased by 7.2%, and the prevalence of moderate-severe depression increased by 10.2%. within four months after the pandemic began in a sample of 419 US first-year students [15].

The most frequently used strategies to cope with COVID-19 in Polish students of 17 universities were acceptance, planning, and seeking emotional support that was less adequate in younger students [16]. Approximately half of the students were found in other studies in need of help to cope with the situation created by the pandemic [10].

The mental health of medical students has been even more concerning compared to their same-aged peers in non-medical courses [17]. The COVID-19 pandemic compounded the mental burden inherent in the medical educational environment [18] while social distancing rules reduced available support [19]. This necessitated the development of novel online support services around the world [20], including Hungary.”

R1.2. You should improve ‘P’ to ‘p’. You should use a. Of course, throughout the text, but only in relation to the level of statistical significance.

BO: ‘P’ were replaced by ‘p’ throughout the text.

R1.3. You should add the subsection ‘Objectives’ or ‘Aims’, in which you should formulate the research questions and/or hypotheses.

BO: We prepared the manuscript following the instructions for authors of IJERPH which does not require separate ‘Objectives’ or ‘Aims’ sections. Aims must be described at the end of the Introduction which we complied with. The aims of our study were as follows:

“In line with recommendations made early on during the pandemic to research its mental health consequences and potential interventions [21], and considering the dearth of research on the patterns of use of e-mental health services [6], by medical students, we aimed at investigating the access to our newly developed, closed-access online mental health support service and the characteristics of those who used the service between 15 April 2020 and 15 April 2021 based on data obtained from the system.”

R1.4. Why did you use the Mann-Whitney test? Could you explain it? This is a nonparametric test.

BO: The nonparametric Mann-Whitney test was used because the continuous dependent variables were not normally distributed, hence the assumptions of the parametric two-samples T-test were not met. A brief explanation is added in the Methods chapter of the manuscript to justify the use of this test:

„Continuous variables due to their non-normal distribution were described by median and interquartile range compared by Mann-Whitney test.”

R1.5. Another very important issue is that the limitations of the study should be formulated.

BO: We extended the limitations of the study described in the Discussion as follows:

„One limitation of the study is the fact that all students were aware of their activities being automatically registered on the portal; this may have been a deterrent for those students who did not want to reveal their help-seeking behaviour. Another limitation is the fact that 15% of the invited students used the portal but this has been in line with other data regarding access to online resources as described above. Generalization of the preferences for interventions to all medical students at the university is not possible due to the limited access but it has been a well-known shortcoming of all preventive methods which target a wide population that many of those who would benefit from interventions do not feel the need to access them.”

Reviewer 2 Report

On a basis of the received findings, I would be beneficial, If the authors could formulate appropriate recommendations for Community action to deal with the stated Educational problems.   

Author Response

Dear Reviewer,

Thank you for your comment which helps us to raise the quality of our manuscript. Please see our response below.

R2.1. On a basis of the received findings, it would be beneficial, if the authors could formulate appropriate recommendations for Community action to deal with the stated Educational problems.

BO: Thank you for the suggestion. Considering the fact that the greatest interest was shown in learning methods, we have extended the ‘Conclusions’ with recommendations for interventions to reach a larger segment of the community of students as follows:  

“Thus, interventions to aid students in their studies seem to be a frequent need that requires problem-focused support. A variety of methods could be offered such as training to improve learning skills that should be part of the mandatory curriculum. Effective training should be delivered by trained professionals for small groups of students in the form of interactive exercises monitoring effectiveness. Peer mentoring can be established by recruiting senior students and pairing them with juniors. Voluntary peer mentoring programs cannot only help students to develop their study skills, but they can also help extend peer relations and integration into the community of medical students.”

Round 2

Reviewer 1 Report

Congratulations!